# Fibrinogenolysis in Venom-Induced Consumption Coagulopathy after *Viperidae* Snakebites: A Pilot Study

**DOI:** 10.3390/toxins14080538

**Published:** 2022-08-06

**Authors:** Jiri Valenta, Alzbeta Hlavackova, Zdenek Stach, Jana Stikarova, Marek Havlicek, Pavel Michalek

**Affiliations:** 1Department of Anaesthesiology and Intensive Medicine, Toxinology Center, First Medical Faculty, Charles University, General University Hospital, U Nemocnice 499/2, 128 08 Prague, Czech Republic; 2Department of Biochemistry, Institute of Hematology and Blood Transfusion, U Nemocnice 2094/1, 128 00 Prague, Czech Republic

**Keywords:** snakebite, *Viperidae*, venom-induced consumption coagulopathy, fibrinogen, western blot

## Abstract

Envenomations that are caused by *Viperidae* snakebites are mostly accompanied by venom-induced consumption coagulopathy (VICC) with defibrination. The clinical course of VICC is well described; however, reports about its detailed effects in the hemocoagulation systems of patients are sparse. In this pilot study, we prospectively analyzed the changes in plasma fibrinogen that were caused by the envenomation of six patients by five non-European *Viperidae* snakes. Western blot analysis was employed and fibrinogen fragments were visualized with the use of specific anti-human fibrinogen antibodies. All of the studied subjects experienced hypo- or afibrinogenemia. The western blot analysis demonstrated fibrinogenolysis of the fibrinogen chains in all of the cases. Fibrinogenolysis was considered to be a predominant cause of defibrination in *Crotalus*, *Echis*, and *Macrovipera* envenomation; while, in the cases of VICC that were caused by *Atheris* and *Calloselasma* envenomation, the splitting of the fibrinogen chains was present less significantly.

## 1. Introduction

Envenomation, by snakebite, from most non-European viperids (family *Viperidae*, subfamilies *Viperinae* and *Crotalinae*) causes, among other symptoms (e.g., gastrointestinal disturbances, local edema and tissue damage, hypotension potentially leading to shock, and kidney injury), venom-induced consumption coagulopathy (VICC). This type of coagulopathy decreases plasma fibrinogen (FBG) levels significantly and leads to the presence of fibrin/fibrinogen degradation products (FDP). VICC may ultimately lead to bleeding or thrombotic microangiopathy and, rarely, it may lead to thrombotic complications [1,2,3]. 

Defibrination in VICC is caused by proteolytic enzymes, such as thrombin-like enzymes (including serine proteases and metalloproteases), or the activators of factors FII, FVII, or FX [4,5].

This subsequently causes the cleavage of the fibrinogen (FBG) chains and eventually the formation of fibrin without stabilization by the factor XIII (FXIII) [3,5,6]. Following this, non-cross-linked forms of FDP are the products of the FBG degradation. Another mechanism leading to consumption coagulopathy is the direct activation of prothrombin and other components of the coagulation cascade, such as factors V, VIII/IX, or X. The result of this mechanism is fibrin formation, including its potential stabilization by the factor XIII with subsequent degradation to cross-linked forms of FDP/D-dimers, which may be the first sign of incoming VICC [7]. FBG and fibrin cleavage are potentiated by the activation of plasminogen [4,7,8]. 

A detailed description of the enzymes’ properties and their effects on coagulation factors in vitro in different snakes has been published elsewhere, mainly with emphasis on the analysis of the venom’s components [9,10,11,12]. The main interest of our pilot study is the changes and dynamics of FBG levels, its breakdown, and reconstitution, in the context of changes in the other hemocoagulation parameters. Each change in FBG on the western blot can be compared with laboratory results and vice versa. Therefore, the use of western blot analysis may broaden our understanding of what occurs during VICC in humans. 

The aim of this prospective pilot study is to analyze the laboratory course of VICC following envenomation by *Viperidae* snakes, with a special emphasis on changes in FBG. The course of ongoing fibrinogenolysis and return to physiological FBG levels (the dynamics of the FBG changes) were monitored over time using western blot analysis wherein the chains of FBG and its fragments were visualized by specific anti-human fibrinogen antibodies and compared with the laboratory parameters of hemocoagulation.

## 2. Results

All of the patients who were included in this pilot study had their hemocoagulation parameters evaluated throughout the course of their envenomation. Special emphasis was put on the total levels of fibrinogen in the blood, these levels’ changes over time, the mode of the fibrinogen’s degradation, and the presence of specific fibrinogen degradation chains. 

Furthermore, in all of the patients, the international normalized ratio (INR) of the prothrombin time (PT), activated partial thromboplastin time (APTT), thrombin time (TT), antithrombin III activity (ATIII), D-dimer levels, and platelet count (PLT) were evaluated at regular intervals. 

### 2.1. Western Blot Analysis of Fibrinogen

In order to monitor the effect of various snake venoms on human fibrinogen, we studied plasma, from different patients, that was submitted to SDS-PAGE under reduced conditions and western blotting reactions using specific antibodies. We found that, in samples that were taken from 12 to 28 h after an *Atheris nitschei* bite, the patient’s plasma contained a slightly decreased intensity of FBG Aα, Bβ, and γ chain bands. The intensity of the Aα chain band and especially that of the γ chain band were visibly increased with prolonged time since the snakebite. In addition, at 12 h since the snakebite, the FBG fragments 27 kDa, 41 kDa, and 44 kDa appeared in comparison with the human fibrinogen standard and the intensity of these bands decreased gradually with time and the increasing intensity of the FBG chains (Figure 1a). Envenomation peaked in 12–18 h since the bite with the lowest FBG and high D-dimer levels (Table 1).

In the case of envenomations by *Calloselasma rhodostoma*, we found that the intensity of the FBG Aα and Bβ chain bands decreased in the patient’s plasma samples 8.5 h after the snakebite. Minimal changes were seen in the γ chain band during the monitoring. FBG fragments 27 kDa and 44 kDa appeared at all of the measured time points after the snakebite in comparison to the human fibrinogen standards. Moreover, with prolonged time since the bite, the intensity of the FBG Aα chain bands increased, especially at 26 h after the snakebite (Figure 1b). The antivenom was administrated 17 h after the envenomation (Table 2).

In the first case of envenomation by *Crotalus adamanteus*, we found that 3.5 and 8 h after the snakebite the patient’s plasma samples did not contain visible bands of FBG Aα and Bβ chains. The intensity of the γ chain had significantly decreased 8 h after the snakebite. In samples from 3.5 h to 26 h after the snakebite, the FBG fragments in the 35–40 kDa and 100–140 kDa regions also completely disappeared. The most intensive FBG fragment 44 kDa and the other 14 kDa, 25 kDa, 27 kDa, and 41 kDa fragments appeared 3.5 h after the snakebite. The FBG Aα, Bβ, and γ chain bands reappeared in the patient’s plasma samples from 16 h post-bite and after the administration of the antivenom. They remained at the same intensity for up to 26 h after the snakebite. The next significant changes occurred 26 h after the bite, following additional antivenom and plasma administration (Table 3). From this time point, we observed increased intensity of the FBG Aα, Bβ, and γ chain bands. FBG fragments in, approximately, the 120, 140 kDa and 35–40 kDa regions appeared. On the contrary, the 14 kDa, 25 kDa, 27 kDa, 41 kDa, and 44 kDa fragments gradually disappeared with prolonging time (Figure 1c). In the second case of *Crotalus adamanteus* envenomation (Table 4), we determined that, at 4.5 h since the snakebite, the patient’s plasma samples did not contain visible bands of the FBG Aα chain. The intensity of the Bβ chain and γ chain bands were significantly decreased. The FBG fragments in the 25–35 kDa region and fragments 41 kDa and 44 kDa appeared in the first sample, i.e., 4.5 h after snakebite. Moreover, with prolonging time, the intensity of the FBG Aα chain rose, especially 19 h after the snakebite. Similarly, the intensity of the FBG fragments in the 25–35 kDa region slightly decreased and the fragments 41 kDa and 44 kDa disappeared with prolonged time since the snakebite (Figure 2a). 

At only 0.5 h after the *Echis coloratus* bite, the samples contained decreased intensities of the FBG Aα chain bands. A large number of the FBG fragments in the 35–50 kDa region and the fragments 12 kDa, 15 kDa, 25 kDa, and 27 kDa appeared in comparison with the human fibrinogen standard. The intensity of the FBG Bβ and γ chain bands was decreased. We observed no visible FBG Aα chain bands nor 120 kDa or 200 kDa bands in the patient’s plasma samples 2.5 and 5.5 h after the bite. We also found a significant decrease in the FBG Bβ and γ chain bands’ intensities. Moreover, intensive FBG fragments 44 kDa and 105 kDa were revealed. The intensity of the FBG fragments decreased with antivenom administration and prolonging time (Table 5). The FBG Aα chain bands, 120 kDa and 200 kDa bands were visible again 12.5 h after the snakebite (Figure 2b). 

In the case of envenomation by *Macrovipera schweizeri*, we found that, 5.5 h after the bite, the patient’s plasma samples contained no visible FBG Aα chain and a significantly decreased intensity of the Bβ and γ chain bands (Figure 2c). The FBG fragments 12 kDa, 15 kDa, 25 kDa, 27 kDa, and 44 kDa appeared. The intensity of these bands decreased with prolonged time. We observed no visible 120 kDa and 200 kDa bands, however these reappeared 34.5 h after the snakebite.

### 2.2. Changes in Fibrinogen Levels

A significant decrease in the FBG levels was present in all of the studied subjects. The changes in the FBG levels in individual patients over time are expressed in Table 1, Table 2, Table 3, Table 4, Table 5 and Table 6. The most significant decreases were recorded in both of the envenomations that were caused by *Crotalus adamanteus*, wherein the first case showed FBG levels that were lower than 0.01 g/L at 20 and 24 h after the snakebite (Table 3) and the second patient demonstrated immeasurable FBG levels in the interval between 4.5 and 10 h post envenomation (Table 4). Only slightly higher minimum levels of FBG were found in cases of *Calloselasma rhodostoma* envenomation—0.05 g/L at 17 h (Table 2) and *Echis coloratus* envenomation—less than 0.1 g/L–at 2.5 h after the snakebite (Table 5). In the case of *Atheris nitschei* envenomation, the lowest measured levels of FBG were 0.2 g/L at 18 h after the bite (Table 1). The decrease in the FBG levels was the least expressed in *Macrovipera schweizeri* envenomation—the lowest value was measured as 1.31 g/L at 5.5 h after the snakebite (Table 6).

**Table 4 toxins-14-00538-t004:** Results of hemocoagulation tests (for *Crotalus adamanteus* case 2).

Hours Since Bite	4.5	6.5	10	19	25
PT/INR	1.61	>10	6.22	1.22	1.11
APTT (s)	31.1	>180	>180.0	27.9	26.9
TT (s)	52.4	>180	>180.0	18.3	15.8
AT III (% activity)	89	95	87	87	96
FBG Claus (g/L)	<0.01	<0.01	<0.01	0.24	1.02
D-dimers (μg/L)	4550	6470	9874	5019	3414
PLT (×10^9^/L)	NA	NA	NA	234	NA
Antivenom (vials)		3			
Sample No.	1	2	3	4	5

**Table 5 toxins-14-00538-t005:** Results of hemocoagulation tests (for *Echis coloratus*).

Hours Since Bite	0.5	2.5	5.5	12.5	18	25	30	42
PT/INR	1.23	>10	>10	1.55	1.33	1.07	1.07	1.03
APTT (s)	45.8	>180	>180	34.2	29.2	24.9	24.3	25
TT (s)	23.4	>180	>180	32.6	NA	18.6	17	15.7
AT III (% activity)	85	90	93	83	NA	93	90	90
FBG Claus (g/L)	0.48	<0.10	0.1	0.18	0.41	0.81	1.62	2.07
D-dimers (μg/L)	13,431	10,351	10,615	11,505	NA	16,440	8834	4428
PLT (×10^9^/L)	225	182	244	198	201	NA	166	155
Antivenom (vials)		3	3					
FFP TU			3					
Sample No.	1	2	3	4	5	6	7	8

**Table 6 toxins-14-00538-t006:** Results of hemocoagulation tests (for *Macrovipera schweizeri*).

Hours Since Bite	5.5	9.0	13.0	17.0	21.0	28.5	34.5	40.5
PT/INR	1.47	1.39	1.23	1.16	1.14	1.12	1.16	1.13
APTT (s)	37	39	36.2	33.2	32.6	34.3	36.6	31.6
TT (s)	23	21.1	17.4	15.3	13.4	13.9	13.9	13.3
AT III (% activity)	62	65	69	62	64	67	63	69
FBG Claus (g/L)	1.31	1.32	1.75	2.17	2.45	2.41	2.36	2.53
D-dimers (μg/L)	8217	>6400	>6400	>6400	5189	2314	1438	1276
PLT (×10^9^/L)	88	NA	NA	46	39	30	21	48
PLT (TU)							1	
Sample No.	1	2	3	4	5	6	7	8

### 2.3. Changes in Other Hemocoagulation Parameters

The coagulation times for PT (INR), APTT, and TT were most affected (>10, >180 s, respectively) following envenomations by *Crotalus* and *Echis* vipers, which corresponded with an extreme decrease in FBG levels and also with the findings of the western blots. The FBG decline was not accompanied by a significant prolongation of the coagulation times in *Atheris*, *Calloselasma,* and *Macrovipera* envenomations. D-dimer was elevated in all of the cases; the highest levels were found following *Atheris* and *Echis* bites. The platelet count was affected only in cases of *Calloselasma* and *Macrovipera* envenomation, wherein they decreased to 44 × 10^9^/L and 21 × 10^9^/L, respectively. A significant decrease in AT III activity was not recorded. The lowest levels of activity, 62% and 55%, were registered in one examination of *Macrovipera* and the *Crotalus* No. 1 case. The sample number in the tables corresponds with the sample number in the western blots; the picture of the FBG degradation process is thus possible to compare with the results of the conventional laboratory analysis of hemocoagulation (Table 1, Table 2, Table 3, Table 4, Table 5 and Table 6).

### 2.4. Clinical Course

Regarding the clinical course of envenomation, we recorded local tenderness, peripheral edema reaching to the elbow or arm, mild subcutaneous hematoma at the bite site, and enlargement of the axillary lymph nodes. One patient, after *Crotalus adamanteus* envenomation, developed a massive hemorrhagic bulla (Figure 3) at the bite side. The systemic presentation of the envenomation manifested as gastrointestinal symptoms, temporary hypotension, and anaphylactic reaction. Serious clinical signs of hemocoagulation disturbances, such as significant bleeding or (micro)embolization, were not observed. The length of the patients’ stays in the intensive care unit were 2–3 days, with a mean of 2.8 days. Antivenom treatment was used in all of the cases, except for those concerning *Atheris* envenomation, because such an antivenom is not produced. In the cases of *Crotalus adamanteus* bites, Antivipmyn Tri, Bioclon, Toriello Guera, Mexico (undetermined whether specific or paraspecific) was used; in *Calloselasma rhodostoma* bite cases Malayan Pit Viper Antivenin, Thai Red Cross, Pathumwaw, Thailand (specific) was used; and in *Echis coloratus* bite cases EchiTAb, Instituto Clodomiro Picado, San Jose, Costa Rica (specific) was administered. In some cases, fresh frozen plasma (FFP) and PLT were administered (Table 1, Table 2, Table 3, Table 4, Table 5 and Table 6). No patient died or suffered permanent damage as a consequence of their envenomation. 

## 3. Discussion

The most significant changes in the results of the western blot analysis following SDS-PAGE under reduced conditions were observed when comparing the human fibrinogen standard with patients’ plasma samples after envenomation by both *Crotalus adamanteus* and *Echis coloratus* bites. In these cases, we found no visible, minimally visible, or significantly decreased intensity of the FBG chain bands. In the first case of *Crotalus adamanteus* envenomation, the experimental data showed the activities of the venom on fibrinogen, causing degradation of the Aα and Bβ chains within 3.5 h post snakebite. Moreover, INR/PT, APTT and TT were not recordable and the FBG concentration was low or undetectable at this time. On the contrary, the γ chain seemed to be more resistant to digestion as the intensity of the γ chain band was only decreased. 

The reason for the disappearance or significant decrease of the individual FBG chain bands after *Crotalus adamanteus* envenomation is their probable cleavage by the enzymes that are contained in the snake venom. Snake venom serine proteases (SVSPs) play a key role in coagulation or fibrinolytic/fibrinogenolytic activities. SVSPs could release either fibrinopeptide A or B with specificity depending on the snake species. In addition, they do not activate FXIII, possibly because they tend to degrade it [13]. The release of fibrinopeptides and no activation or degradation of FXIII result in non-cross-linked fibrin formation [14]. Thrombin-like enzymes (TLEs) are capable of limited cleavage of fibrinopeptides A and B and the conversion of fibrinogen to fibrin. Unlike thrombin, they activate FXIII or platelets in only a minimum of cases. A major part of TLEs is formed by the zinc metalloproteinases that preferentially cleave the Aα chain of fibrinogen. The second part of TLEs is formed by serine protease, which has specific activity toward the Bβ fibrinogen chain [4]. TLEs simply consume fibrinogen rather than activating the clotting pathway [5]. Both of these effects lead to fibrinogen consumption due to fibrinogen degradation without the conversion to fibrin [15]. Fibrinolytic activities, by the means of the direct degradation of fibrinogen/fibrin and plasminogen activation that are caused by snake venoms, lead to a decrease in FBG [1,16,17]. This finding agrees with those of the other measured parameters whose results were unrecordable, including INR/PT, APTT, TT, and low FBG concentration. Fibrinogen consumption that is induced by TLEs results in an unrecordable PT/INR and bleeding complications if there is an undetectable level of FBG. This has been confirmed in our cases. In the western blot analysis, TLEs caused no visible FBG chain bands and moreover, the cleavage of major Aα and Bβ chains resulting in the destruction of FBG without its conversion to fibrin. The evidence of the TLEs cleavage of the FBG chains is also supported by the presence of FBG fragments in the 40–45 kDa region which could have arisen as a fragment of the alpha chain’s cleavage. Paes Leme et al. reported, in their work, about the 44 kDa FBG fragment which was shown to originate from the fibrinogen alpha chain, not fibrin [18]. A 41 kDa fragment could originate from the γ′ chain of FBG [19]. Fragments of similar sizes to the ones that we found (i.e., the 14 kDa, 25 kDa, and 27 kDa fragments) have been described in papers as a result of fibrin(ogen) cleavage by plasmin and the effect of rattlesnake venom. The 14 kDa and 25 kDa fragments could probably also originate from the α chain and the 27 kDa from the γ chain [20,21]. This assumption is also confirmed by the fact that these FBG fragments disappear over time with the increasing intensity of FBG chain bands. On the contrary, some of the bands of FBG fragments disappeared after the bite and reappeared as time increased after the snakebite, these are FBG fragments at 120 kDa and 140 kDa. These fragments could be the result of FBG’s covalent cross-link generation with other plasma proteins as reported by Mosesson et al. [22]. The only issue that is remaining is the increase in the concentration of D-dimer. D-dimer is a marker of fibrin degradation after the polymerization of FBG to a normally formed cross-linked fibrin clot. High levels of D-dimer correspond to the possibility that fibrinogen is also converted to non-cross-linked fibrinogen degradation products (FDPs) by the venom. The probable explanation is due to the use of a less specific D-dimer assay which was, in our case, distorted by the high concentration of interfering fibrinogen cleavage and/or degradation products [5]. This statement is also supported by the negligible concentration of D-dimer that was found compared to the theoretical plasma concentration of FBG and the absence of (micro)thrombi clinical symptoms. Therefore, in *Crotalus adamanteus* envenomation cases, all of the indications suggest that fibrinogenolysis predominates over the activation coagulation cascade.

Toxins of *Echis coloratus* have a mechanism causing coagulopathy that is similar to the mechanism of prothrombin activators [5]. *Echis* species contain metalloproteinase prothrombin activator toxins that not only activate the clotting pathway but also act as hemorrhagins. Thus, these metalloproteases directly activate prothrombin but they convert it into the less active meizothrombin, rather than the fully active thrombin [13]. Finally, many snake venom metalloproteases are capable of digesting fibrinogen/fibrin directly [14,23]. The second activity is the degradation of the extracellular matrix and vascular basement membrane following damage of blood vessel wall. Fibrinogenolysis was apparent and contributed significantly to the defibrination that was observed in both of the cases of *Crotalus adamanteus* envenomation, as well as in the case of *Echis coloratus* envenomation, despite the fact that the venom contains prothrombin activators [5]. The origin of the FBG fragments 25 kDa, 27 kDa, 44 kDa, and 120 kDa was discussed in the part of this paper that discusses *Crotalus adamanteus* envenomation. The 12 kDa fragment could be released from the non-crosslinked fibrin γ chain resp. the 15 kDa fragment from the fibrin α chain [24]. FBG fragments in 35–50 kDa regions were generated intensively due to the snake envenomation and their origin has not yet been described. The observed 200 kDa fragment could be generated as a multimer of the FBG alpha chains [25]. A similar finding of a decrease in the FBG chains and FBG fragments was also shown by western blot analysis in the case of *Macrovipera schweizeri* envenomation. 

We found that the patients’ plasma samples contained a slightly decreased intensity, but not disappearance, of the FBG Aα, Bβ, and γ chains at 12 h after an *Atheris nitschei* bite. With prolonged time, the visibility of the fibrinogen chains increased and that of the fibrinogen fragments 41 kDa and 44 kDa decreased. The 27 kDa fragment was probably generated from the γ chain and only slightly decreased as the γ chain’s intensity increased [21]. The origin of the other FBG fragments was described in part in the section of this paper addressing *Crotalus adamanteus* envenomation. The venom components of the genus *Atheris* include TLEs that simply consume FBG rather than activating the clotting pathway [7]. The procoagulant activity was observed in *Atheris squamiger* due to activation via factor V [26]. However, in the reported case, procoagulant activity was found to be negligible and the main role in afibrinogenemia was credited to fibrinogen–convertase activity. In our case, AT III was also in the normal range without any dramatic changes; as such there is the assumption that thrombin is not activated in surplus. 

TLEs cause an isolated deficiency of fibrinogen. Observed hypofibrinogenemia is the main cause of prolonged time in coagulation tests. However, western blot analysis suggests that the fibrinogenolysis was less intensive compared to the measured concentration of D-dimer as a degradation product of cross-linked fibrin in this case. In the case of a *Calloselasma rhodostoma* bite, similar FBG fragments were found by western blot analysis. The origin of these fragments is probably the same as that which has been discussed above.

The main limitation of our study is the small range of patients with different treatments, which could affect the observed parameters as well as the generation of the FBG fragments that are visible using western blot analysis. The use of western blot analysis followed by SDS-PAGE under reduced conditions did allow us to differentiate between fragments but their origins could not be verified. This method provides us with a visual description of the course of fibrinogen consumption over time as a result of a snakebite. A small number of studied envenomations also does not allow for statistical analysis. Further studies, ideally in collaboration with other toxinology centers in Europe and worldwide, are needed in order to confirm our findings. Another limitation is that some of the timings of the blood samplings differed between the cases. This was caused, firstly, by different admission times to the ICU after the bite and, secondly, by the requirements for blood samplings following therapeutic interventions such as antivenom administration or fresh frozen plasma application. 

## 4. Conclusions

The results of this pilot study have confirmed that fibrinogenolysis leading to hypofibrinogenemia is the main finding in VICC following *Viperidae* snakebites. The interpretation of the results of this pilot study is limited due to the low number of subjects. Western blot analysis has illustrated the course of fibrinogenolysis and offers further information about the causes of hypofibrinogenemia in VICC. The changes that were observed in our study should be also confirmed by experiments with the use of purified fibrinogen and its reaction to different *Viperidae* venoms. 

## 5. Materials and Methods

Following Hospital Ethical Committee approval—number 2361/15 S-IV (approved 10 December 2015 updated 16 January 2018. Chair: Dr. Josef Sedivy)—we prospectively included in this pilot study six patients who developed VICC following snakebites by non-European *Viperidae* snakes: the Great Lake bush viper *Atheris nitschei*, Malayan pit viper *Calloselasma rhodostoma*, Eastern diamond-backed rattlesnake *Crotalus adamanteus* (two cases), Palestine saw-scaled viper *Echis coloratus*, and Cyclades blunt-nosed viper *Macrovipera schweizeri*. The patients were four amateur and two professional snake-breeders who were aged 31–67 years (mean: 46 years). They did not have any significant co-morbidities in their history.

First, blood samples for laboratory evaluation were taken 0.5–8.75 h (mean: 5.8 h; median: 4 h) after the bite. We administered the antivenom to four out of the six patients (in total 3–11 doses). The first dose of the antivenom was applied in the range of 2.5–17 h after the snakebite. Antivenom for the treatment of *A. nitschei* and *M. schweizeri* envenomation does not exist.

All of the patients were, following the appearance of the clinical or laboratory signs of envenomation in district hospitals, transferred for complex treatment to the specialized Toxinology Center for the Czech Republic, located at the Department of Anesthesiology and Intensive Medicine, General University Hospital, Charles University in Prague, Prague, Czech Republic.

The blood samples were taken for the laboratory evaluation of the hemocoagulation parameters during the course of the envenomation and treatment. A part of the plasma samples was deeply frozen to the temperature of −80 °C and subsequently stored for the further processing of western blot analyses. Fibrinogen changes were observed and fibrinogen fragments were visualized by the use of specific anti-human fibrinogen antibodies. 

The western blot analysis can be described as follows: the FBG standard of concentration was 1 g/L (Sigma-Aldrich, Prague, Czech Republic) and the plasma samples were diluted 10 times with PBS. The samples, for denaturing protein gel electrophoresis on NuPAGE^TM^ Bis-Tris Mini Gels, were prepared under reducing conditions according to the NuPAGE technical guide (ThermoFisher Scientific, Life Technologies Czech Republic Ltd., Prague, Czech Republic). Briefly, 4 µL of the diluted sample was mixed with 5 µL NuPAGE LDS Sample Buffer (4X), 2 µL NuPAGE Reducing Agent (10X), and 9 µL H_2_O. The samples were heated at 70 °C for 10 min. The plasma protein samples were separated by SDS-PAGE in 4–12% gradient NuPAGE^®^ Bis-Tris Mini gel using 1× NuPAGE morpholinepropanesulfonic acid running buffer according to the manufacturer’s instructions. The separated proteins were transferred onto polyvinylidene fluoride (PVDF) membranes using Owl™ HEP Series Semidry Electroblotting Systems (10 V, 80 mA for 30 min and 10 V, 200 mA for 90 min). The non-specific binding sites were blocked by 5% nonfat milk in 0.1% Tween/Tris-buffered saline for 1 h at room temperature. The FBG immunodetection was performed using rabbit anti-human fibrinogen polyclonal antibody (DAKO, HPST, Prague, Czech Republic; dilution 1:50,000 in 0.1% Tween/Tris-buffered saline) for 1 h at room temperature. Then, the blots were washed three times with 0.1% Tween/Tris-buffered saline and incubated with goat anti-rabbit antibody that was conjugated with alkaline phosphatase (Sigma-Aldrich, Prague, Czech Republic; dilution 1:50,000 in 0.1% Tween/Tris-buffered saline) for 1 h at room temperature. The visualization of the immunoreactions was carried out by using the BCIP/NBT liquid substrate system reaction (Sigma-Aldrich, Prague, Czech Republic).

## Figures and Tables

**Figure 1 toxins-14-00538-f001:**
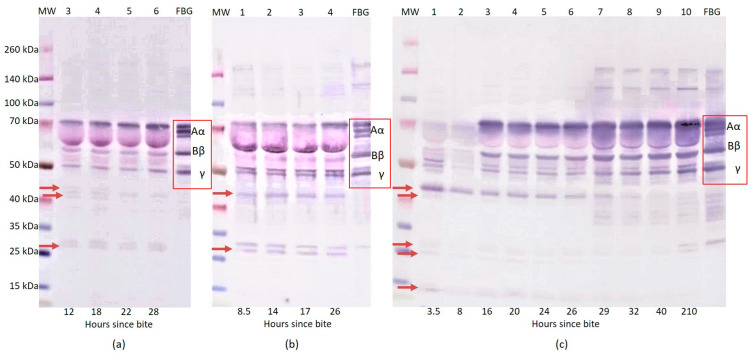
Results of western blot analysis of patient’s plasma proteins after snakebite, over time. Plasma proteins were separated by SDS-PAGE under reduced conditions followed by western blot analysis using polyclonal anti-human fibrinogen antibodies (DAKO, HPST, Prague, Czech Republic). Western blot analysis of human fibrinogen in patients’ plasma after bites by *Atheris nitschei* (**a**), *Calloselasma rhodostoma* (**b**), *Crotalus adamenteus* case 1 (**c**). MW—protein molecular weight marker (Thermo Scientific^TM^, Spectra^TM^ Multicolor Broad Range Protein Ladder, Life Technologies Czech Republic Ltd., Prague, Czech Republic). Molecular masses (kDa) are indicated on the left side. FBG—human fibrinogen standard. The sample number is provided in the top line and hours since bite in the bottom line. Red arrows mark fibrinogen fragments induced by snakebite. The chains of human fibrinogen standards are highlighted with red square boxes. Note: Samples 1 and 2 in (**a**) are missing. They were not analyzed by Western blot.

**Figure 2 toxins-14-00538-f002:**
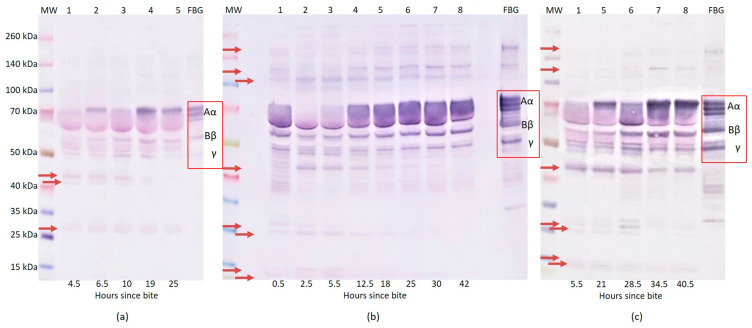
Results of western blot analysis of patients’ plasma proteins after snakebite over time. Plasma proteins were separated by SDS-PAGE under reduced conditions followed by western blot analysis using polyclonal anti-human fibrinogen antibodies (DAKO, HPST, Prague, Czech Republic). Western blot analysis of human fibrinogen in patients’ plasma after bites by *Crotalus adamanteus* case 2 (**a**), *Echis coloratus* (**b**), *Macrovipera schweizeri* (**c**). MW—protein molecular weight marker (Thermo Scientific^TM^, Spectra^TM^ Multicolor Broad Range Protein Ladder, Life Technologies Czech Republic Ltd., Prague, Czech Republic). Molecular masses (kDa) are indicated on the left side. FBG—human fibrinogen standard. The sample number is provided in the top line and hours since bite in the bottom line. Red arrows mark fibrinogen fragments induced by snakebite. The chains of human fibrinogen standards are highlighted with red square boxes.

**Figure 3 toxins-14-00538-f003:**
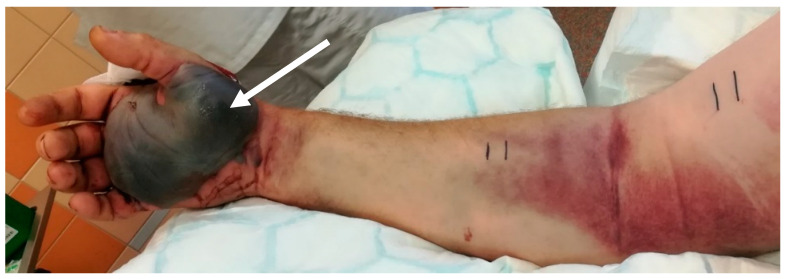
*Crotalus adamanteus* (case 1): Hemorrhagic bulla (white arrow) at the bite site, 22 h post-bite.

**Table 1 toxins-14-00538-t001:** Results of hemocoagulation tests (for *Atheris nitschei*). Legend for the tables: abbreviations: PT/INR—prothrombin time/the international normalized ratio; APTT—activated partial thromboplastin time; TT—thrombin time; s—seconds; AT III—antithrombin III; FBG—fibrinogen; PLT—platelets; NA—not applicable.

Hours Since Bite	0.75	3.75	12	18	22	28	36
PT/INR	0.94	1.02	1.13	1.38	1.35	1.31	1.17
APTT (s)	26.1	27.8	27.8	36.8	34.8	33.2	32.8
TT (s)	NA	NA	NA	22.8	19.4	16.8	15.0
AT III (% activity)	105	84	74	81	76	77	81
FBG Claus (g/L)	2.6	1.6	0.3	0.2	0.3	0.3	0.94
D-dimers (μg/L)	210	3110	18,500	5190	2226	1100	691
PLT (×10^9^/L)	186	NA	159	180	172	172	180
Sample No.	1	2	3	4	5	6	7

**Table 2 toxins-14-00538-t002:** Results of hemocoagulation tests (for *Calloselasma rhodostoma*).

Hours Since Bite	8.5	14	17	26	33	38	44	50
PT/INR	1.15	1.35	1.68	1.43	1.49	1.38	1.28	1.23
APTT (s)	29.1	33.6	38.2	31.4	36.8	35.4	33.8	30.6
TT (s)	31.7	46.3	60.1	45.4	33.5	24.4	20.5	18.6
AT III (% activity)	89	89	89	81	85	88	87	80
FBG Claus (g/L)	0.8	0.14	0.05	0.11	0.2	0.23	0.59	0.65
D-dimers (μg/L)	2616	4526	5096	2176	890	575	85	476
PLT (×10^9^/L)	50	45	44	142	NA	130	NA	121
Antivenom (vials)			3					
Sample No.	1	2	3	4	5	6	7	8

**Table 3 toxins-14-00538-t003:** Results of hemocoagulation tests (for *Crotalus adamanteus* case 1).

Hours Since Bite	3.5	8	16	20	24	26	29	32	40	210
PT/INR	>10	>10	3.89	4.41	4.44	2.38	1.27	1.18	1.14	1.1
APTT (s)	>180	>180	56.5	56.9	62	42.1	31.9	32.5	27.2	31.7
TT (s)	>180	>180	25.2	25	33.6	26	16.5	16.8	14.4	12.7
AT III (% activity)	61	58	55	59	55	59	65	66	77	78
FBG Claus (g/L)	0.1	0.16	1.13	<0.01	<0.01	0.1	1.27	1.35	1.65	2.97
D-dimers (μg/L)	7321	4591	4269	5251	2078	1817	633	529	274	378
PLT (×10^9^/L)	84	NA	174	NA	NA	NA	154	NA	130	NA
Antivenom (vials)	4	3			4					
FFP (TU)						6				
Sample No.	1	2	3	4	5	6	7	8	9	10

## Data Availability

The authors confirm that the data supporting the findings of this study are available within the article (Table 1, Table 2, Table 3, Table 4, Table 5 and Table 6).

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
