# Peer review of "Fibrinogenolysis in Venom-Induced Consumption Coagulopathy after *Viperidae* Snakebites: A Pilot Study"

_toxins, 2022, doi:10.3390/toxins14080538_

Round 1
Reviewer 1 Report
Dear colleagues, thank you for giving me the opportunity to review this very interesting paper. This work gives us new insights in bleeding pathophysiology in the context of Viperidae bites.
I have only few comments
-discussion: Atheris venom is supposed to contain a factor V activator (Mebs D et al. Severe coagulopathy after a bite of a green bush viper (Atheris squamiger): Case report and biochemical analysis of the venom. Toxicon. 1998). Therefore activation of coagulation is possible with this venom
- I think you can discuss two other limits of your work. WB analysis seems unable to distinguish direct fibrinogenolysis due to venom enzymes such as TLE or fibrinogenases (Crotalus bite), and thrombin-mediated fibrinogenolysis due to thrombin generation after factor activation (Echis bite). Secondly, first case of C. adamanteus bite et case of E. coloratus received FFP which can influence the spontaneous kinetics of fibrinogenolysis.
- Materials and methods: please indicate name of antivenom and if it is specific or paraspecific against species involved in cases.
Author Response
Reviewer No. 1:
I have only few comments
-discussion: Atheris venom is supposed to contain a factor V activator (Mebs D et al. Severe coagulopathy after a bite of a green bush viper (Atheris squamiger): Case report and biochemical analysis of the venom. Toxicon. 1998). Therefore activation of coagulation is possible with this venom.
R: Procoagulant activity was observed in Atheris squamiger due to activation via F V (Mebs D, Holada K, Kornalik F, Simák J, Vanková H, Müller D, et al. Severe coagulopathy after a bite of a green bush viper (Atheris squamiger): case report and biochemical analysis of the venom. Toxicon 1998; 36: 1333 – 1340.) But in the reported case (Mebs D, Holada K, Kornalik F, Simák J, Vanková H, Müller D, et al. Severe coagulopathy after a bite of a green bush viper (Atheris squamiger): case report and biochemical analysis of the venom. Toxicon 1998; 36: 1333 – 1340.) procoagulant activity was evaluated as negligible and main role in afibrinogenemia was credited to fibrinogen-convertase activity. In our case, AT III is also in normal range without dramatical changes so there is assumption that thrombin is not activated in surplus.
- I think you can discuss two other limits of your work. WB analysis seems unable to distinguish direct fibrinogenolysis due to venom enzymes such as TLE or fibrinogenases (Crotalus bite), and thrombin-mediated fibrinogenolysis due to thrombin generation after factor activation (Echis bite). Secondly, first case of C. adamanteus bite et case of E. coloratus received FFP which can influence the spontaneous kinetics of fibrinogenolysis.
R: Thank you for your suggestion. Other limitations of our study were added to the discussion part of article. Limitations of our study is small range of patients with different treatment which could affect observed parameters as well as generation of FBG fragments visible using western blot analysis. Western blot analysis followed SDS-PAGE under reduced conditions could differentiate between fragments but their origin could not be verified. The method provides us with a visual description of the course of fibrinogen consumption over time as a result of a snake bite.
- Materials and methods: please indicate name of antivenom and if it is specific or paraspecific against species involved in cases.
R: Antivenom treatment was used in 4 cases, except Atheris and Macrovipera, because these antivenoms are not produced. In cases of Crotalus adamanteus we used Antivipmyn Tri, Bioclon, Mexico (undetermined whether specific or paraspecific), in the envenoming by Calloselasma rhodostoma Malayan Pit Viper Antivenin, Thai Red Cross, Thailand (specific) and in Echis coloratus EchiTAb, Instituto Clodomiro Picado, Costa Rica (specific).
Reviewer 2 Report
In this paper, the authors report the phenomenon of fibrinogenolysis by VICC for venomous snakebites of the viper family. As a reviewer, I am interested in discussing the Fibrinogenolysis phenomenon and its mechanism of action, but I hope that the following improvements will be made due to the shortcomings of the paper.
1. The data used in this paper are all 1 case except 2 cases of Crotalus adamenteus. This does not confirm the reproducibility. It is recommended to use data from 3 cases for at least 1 snakebite.
2. As blood coagulation-related data, data on ATIII and PLT are shown, but there is no description on these data, so please include a description.
3. The description of the decomposition products of FBG shown in Figure 1 and 2 is insufficient, so please supplement it.
4. Since only a part of the discussion in the discussion on the decomposition products of FBG shown in Figure 1 and 2 is given, please supplement the discussion including the analysis of each decomposition product.
5. In the discussion, there is no description about the limitation of this paper, so please add the description in separate paragraphs.
6. Authors discuss the possibility of increased D-dm and FDP as the reason for the decrease in FBG by Crotalus adamenteus, but the concentration of D-dm is lower than that of other snakes bite, and FDP is not shown. Readers are hard to convince. Therefore, please add an explanation based on the data.
7. Regarding Minor point, there are two types of descriptions in Table 1-6, "INR / PT" and "PT / INR", but we would like you to unify the notation.
Author Response
Reviewer No. 2:
In this paper, the authors report the phenomenon of fibrinogenolysis by VICC for venomous snakebites of the viper family. As a reviewer, I am interested in discussing the Fibrinogenolysis phenomenon and its mechanism of action, but I hope that the following improvements will be made due to the shortcomings of the paper.
- The data used in this paper are all 1 case except 2 cases of Crotalus adamenteus. This does not confirm the reproducibility. It is recommended to use data from 3 cases for at least 1 snakebite.
R: This is a pilot study to investigate the potential of WB type of FBG analysis, the plasma fibrinogenolysis profile of envenomed patients. In European conditions with a small number of injured venomous snake breeders, the study could not be extended, reproducibility and statistical processing are not possible. Further studies, ideally in collaboration with other Toxinology Centers in other European countries, are needed. We added this text to the Limitations section of the manuscript.
- As blood coagulation-related data, data on ATIII and PLT are shown, but there is no description on these data, so please include a description.
R: Limit values of PLT and AT III have been added to manuscript under 2.3. Changes in other hemocoagulation parameters
„A significant decrease in AT III activity was not recorded in our cohort.
Platelet count was affected only in cases of Caloselasma and Macrovipera envenoming, where decreased temporarily to 44 x109/L and 21 x109/L respectively. A significant reduction in AT III activity was not recorded. Lowest activity 62% and 55% were registered in one examination at Macrovipera and Crotalus No1 cases.“
- The description of the decomposition products of FBG shown in Figure 1 and 2 is insufficient, so please supplement it.
R: Thank you for your comment. For clarity purpose, results were composed as description of the decomposition products of FBG changing with prolonging time since the snake bite. Figure 1 and 2 descriptions were expanded.
„Results of western blot analysis of patients’ plasma proteins after snake bite over time. Plasma proteins were separated by SDS-PAGE under reduced conditions followed by western blot analysis using polyclonal anti-human fibrinogen antibodies (DAKO, HPST, Prague, Czech Republic).“
- Since only a part of the discussion in the discussion on the decomposition products of FBG shown in Figure 1 and 2 is given, please supplement the discussion including the analysis of each decomposition product.
R: Thank you for your comment. Following paragraphs were added to the Discussion section of the manuscript:
The most significant changes in western blot analysis under reduced conditions were observed comparing human fibrinogen standard with patient´s plasma samples after envenomation by both Crotalus adamanteus and Echis coloratus bites.
Thrombin-like enzymes (TLEs) are capable of limited cleavage of fibrinopetides A and B and conversion of fibrinogen to fibrin. Unlike thrombin, they activate FXIII or platelets in minimum cases. A major part of TLEs is formed by zinc metalloproteinases that preferentially cleave Aα chain of fibrinogen. The second part of TLEs is formed by serine protease that has specific activity towards Bβ fibrinogen chain [5]. TLEs simply consume fibrinogen rather than activating the clotting pathway [7].
41kDa fragment could originate from γ´ chain of FBG [odkaz vložit Přidat!]. Fragments of similar size to the ones we found (as 14kDa, 25kDa and 27kDa fragments) have been described in papers as a result of fibrin(ogen) cleavage by plasmin and the effect of rattlesnake venom. 14kDa and 25kDa fragments could probably also originated from α chain and 27kDa from γ chain.
On the contrary, some bands of FBG fragments disappeared after the bite and reappeared as time increases after the snake bite. These are FBG fragments 120kDa and 140kDa. These fragments could be the result of FBG covalently cross-link generation with other plasma proteins as reported by Mosesson et al.
The origin of FBG fragments 25kDa, 27kDa, 44kDa and 120kDa was discussed in part of Crotalus adamanteus envenomation. 12kDa fragment could be released from non-crosslinked fibrin γ chain resp. 15kDa fragment from fibrin α chain [odkaz vložit]. FBG fragments in 35- 50kDa regions were generated intensively due to the snake envenomation and their origin was not described yet. 200kDa fragment could be generated as multimer of FBG alpha chains. A similar finding of a decrease in FBG chains and FBG fragments showen western blot analysis also in the case of Macrovipera schweizeri envenoming.
27kDa fragment was probably generated from γ chain and was only slightly decreasing as γ chain intensity increased. The origin of the other FBG fragments was described in part of Crotalus adamanteus envenomation.
Procoagulant activity was observed in Atheris squamiger due to activation via F V (Mebs D, Holada K, Kornalik F, Simák J, Vanková H, Müller D, et al. Severe coagulopathy after a bite of a green bush viper (Atheris squamiger): case report and biochemical analysis of the venom. Toxicon 1998; 36: 1333 – 1340.). But in the reported case (vložit tento odkaz: Mebs D, Holada K, Kornalik F, Simák J, Vanková H, Müller D, et al. Severe coagulopathy after a bite of a green bush viper (Atheris squamiger): case report and biochemical analysis of the venom. Toxicon 1998; 36: 1333 – 1340.) procoagulant activity was evaluated as negligible and main role in afibrinogenemia was credited to fibrinogen-convertase activity. In our case, AT III is also in normal range without dramatical changes so there is assumption that thrombin is not activated in surplus.
In the case of Calloselasma rhodostoma bite the similar FBG fragments were found by Western blot analysis. The origin of these fragments is probably the same as discussed above.
Limitations of our study is small range of patients with different treatment which could affect observed parameters as well as generation of FBG fragments visible using western blot analysis. Western blot analysis followed the SDS-PAGE under reduced conditions could differentiate between fragments but their origin could not be verified. The method provides us with a visual description of the course of fibrinogen consumption over time as a result of a snake bite.
- In the discussion, there is no description about the limitation of this paper, so please add the description in separate paragraphs.
A: Thank you for your suggestion. Limitations of our study were added to the discussion part of article.
„Limitations of our study is small range of patients with different treatment which could affect observed parameters as well as generation of FBG fragments visible using western blot analysis. Western blot analysis followed the SDS-PAGE under reduced conditions could differentiate between the fragments but their origin could not be verified. The method provides us with a visual description of the course of fibrinogen consumption over time as a result of a snake bite.“
- Authors discuss the possibility of increased D-dm and FDP as the reason for the decrease in FBG by Crotalus adamenteus, but the concentration of D-dm is lower than that of other snakes bite, and FDP is not shown. Readers are hard to convince. Therefore, please add an explanation based on the data.
A: Thank you for your comment.
Possible differences in concentration of D-dimers could be caused by different course of composition of snake venoms and their amount and also by treatment where antivenom vials were used per “lege artis”.
In this study FDPs were not evaluated. D-dimers assays were discussed as not specific for only D-dimers determination and FDP were mentioned as one of possible phenomena leading to false increase of D-dimers.
- Regarding Minor point, there are two types of descriptions in Table 1-6, "INR / PT" and "PT / INR", but we would like you to unify the notation.
A: Thank you for reminder, the abbreviation has been unified to PT/INR.
Reviewer 3 Report
Reference manuscript TOXINS June 2022: “Fibrinogenolysis in Venom-Induced Consumption Coagulopathy after Viperidae Snakebites: A Pilot Study”.
General Comments:
In the manuscript submitted to TOXINS, the authors studied changes in plasma fibrinogen in patients who suffered accidents after being bitten by five different types of snakes (Genus Crotalus, Echis, Macrovipera, Atheris and Calloselasma). The analyzes were performed on the patients' plasma by means of Western Blotting reactions with an antibody that recognizes fibrinogen, and the authors reported that patients injured by snake bites of Crotalus, Echis, Macrovipera had degraded fibrinogen chains, while patients injured by Atheris and Calloselasma snakebites showed less significant fibrinogen degradation. The subject is consistent with the TOXINS line of publications, the text is clear and coherent, the authors have clearly shown their results and in my opinion is that a revised version with minor revision can be accepted for publication. The topic has a great impact on public health in developing countries, where many snakebite accidents are reported daily and these data can help in understanding the mechanistic of post-envenoming pathology.
Comments
1- Between lines 8 and 9 the authors wrote …. we prospectively analyzed changes in plasma fibrinogen caused by the envenoming of five non European Viperidae snakes in six patients. Undoubtedly, the statistics of studying only 6 injured patients is low. This is a criticism that authors should be prepared to respond to! If a larger number of cases were studied, would the results always agree with the data described here?
2- Between lines 22 and 24 the authors wrote … The envenoming by snakebites of most non-European viperids (family Viperidae, sub- family Viperinae and Crotalinae) causes, among others, venom-induced consumption coagulopathy (VICC). Although the authors are studying venom-induced consumption coagulopathy effects after accidents caused by different snakes, in this part of the text they could indicate the other types of clinical effects also described after envenomations! Making it clear that coagulation effects are not the only ones seen after accidents. This can undoubtedly help less experienced clinicians in the field who may have access to the text.
3- Between lines 28 and 29 the authors wrote … Defibrination in VICC is caused by fibrino(geno)lytic enzymes including thrombin- like enzymes. In my opinion, the authors should replace Thrombin-like enzymes by proteolytic enzymes, since some metalloproteases can also degrade plasma substrates, in addition to serine proteases. It would also be more elegant to cite bibliographic references for this sentence!
4- Between lines 38 and 40 the authors wrote … A detailed description of the enzyme properties and their effect on coagulation factors “in vitro” in different snakes has been published elsewhere, mainly with the emphasis on the analysis of venom components. Here, no doubt, the authors could indicate some of these references. This would make the text more elegant and complete, in addition to showing that they (the authors have expertise in the literature of the area)!
5- Throughout the text, the authors describe their results obtained through WB reactions, analyzing the plasmas of different patients with antibodies that recognize fibrinogen. The WB reactions, after standardized by the laboratory, are easy to reproduce and reasonably cheap. However, the authors could have described other inexpensive laboratory and clinical parameters, considering developing countries, such as blood count with platelet counts, physical examinations of patients performed by clinicians with experience in the area, seeking a correlation with the data obtained in the analyzes of plasma fibrinogen and possible streamlining of patient treatments. This could make the described data on fibrinogen degradation after envenomation more complete and attractive to readers in the field.
6- Between lines 61 and 62 the authors wrote … To monitor the effect of various snake venoms on the fibrinogen molecule, we used SDS-PAGE followed by western blot analysis of reduced patient plasma samples. Better rewrite this sentence … To monitor the effect of various snake venoms on the fibrinogen, we studied plasma from different patients submitted to SDS-PAGE under reduced conditions and Western Blotting reactions using specific antibodies.
7- Throughout the text, please substitute molecular weight for molecular mass. Molecules have molecular mass, not molecular weight. Who has weight is rice, bread, meat …. Although this term molecular weight is accepted and used in the literature, the most correct is molecular mass. There is no weight spectrometry, but mass spectrometry!
8- Between lines 64 and 66 the authors wrote... The intensity of the Aα chain bands and especially the γ chain band was visibly increased with prolonging time since snakebite. In my opinion, this diffuse band, with the shape of a fingernail facing downwards, and which is just below the fibrinogen Aα band, seems to be albumin, the most abundant protein in human plasma, and which colocalizes in this region on the SDS-PAGE under reduced conditions (Between 60-67 kDa). As albumin is very abundant in plasma, it gives the impression of a reaction with the anti-fibrinogen antibody, but it appears only due to the high concentration. As a control, the authors should use plasma obtained from uninjured patient and normalized in protein concentration as a reference, and they could see that this band in 67 kDa is albumin using an antibody for instance. It is also likely that the Aα chain has suffered a small degradation, and the product formed after the action of venom proteases now colocalize with Albumin. Giving the impression that the Aα chain has increased, but in reality it has decreased its molecular mass because it has been degraded.
9- Between lines 98 and 99, for SDS-PAGE WB reaction, figure 1a, why lanes started with 3, 4, 5 and 6 instead 1, 2, 3, ….
10- If the authors' objective is really to verify the action of toxins with proteolytic activities found in the crude venoms of different snakes, on fibrinogen, why didn't they study the actions of these crude venoms on purified fibrinogen, rather than on patient plasma? which is a very “contaminated” sample, containing hundreds of other proteins in addition to fibrinogen, and this masks the interpretations of the results.
11- Undoubtedly, the results postulated by the authors of plasma fibrinogen degradation by different snake venoms could be confirmed by the exposure of purified fibrinogen to different venoms. By doing this experiment, the authors could verify which fibrinogen chains are actually degraded by the different venoms and which peptides are generated after fibrinogen degradation. Human plasma has so many proteins that difficult the interpretation of results.
12- In figures 1 and 2, the authors should have followed the same times after the different accidents, for a better comparative analysis. They Need to explain in the revised version, why times were different.
13- Between lines 71 and 73 the authors wrote about figure 1b …. In the case of Calloselasma rhodostoma envenoming, we found that intensity of FBG Aα and Bβ chains bands was decreased in the patient´s plasma samples 8.5 h after snake bite. In my opinion since there are colocalization of Aα chain and albumin this conclusion is masquerade and the authors can revise this conclusion performing this same experiment with purified fibrinogen!
14- Also regarding fig. 1b …. FBG fragments 27kDa and 44kDa appeared at all time points after snakebite in comparison with the human fibrinogen standards. The best comparative analysis in this case was with human plasma of a reference normal donor.
15- In the lines 88 and 89, about figure 1c, the authors wrote … FBG fragments in 100- 140kDa … regions appeared. In a revised version better correct to fragments with approximately 120 and 150 kDa. The authors also need to explain why these same fragments are also present in the FBG used as a reaction control.
16- The WB reaction that describes the immunostaining of fibrinogen in figure 2a is questionable, since even the staining profile of the FBG used as a reference is not good! In this case the conclusions are questioned by the fact that even the control shows practically no marking of the γ chain. In addition, the Aα chain marking profiles also oscillate between lower markings in lanes 1 and 3 and higher markings in lanes 2, 4 and 5.
17- All tables must be accompanied by legends explaining the respective reading times for PT/INR, would it be seconds or minutes?, would the letters (s) after APTT and TT be seconds? Also standardize the writing in the PT/INR or INR/PT in all tables
18- It is good procedure that morphological descriptions have symbols indicating the alterations shown. Thus, in the Figure 3 of the revised version, the authors could place an arrow pointing to the hemorrhagic blister in the injured patient's hand. This may seem very obvious, but it follows a good procedure of morphological and descriptive analysis!
19- At line 176, … The most significant changes in western blot analysis under reduced conditions …change to … The most significant changes in western blot analysis followed a SDS-PAGE under reduced conditions… because electrophoresis that was under reduced conditions and not WB!
20- At lines 178 to 181 the authors wrote ….In these cases, we found no visible, minimally visible, or significantly decreased intensity of FBG chains bands. In the first case of Crotalus adamanteus envenoming, both FBG Aα and Bβ chains bands were not visible 3.5 h post snakebite. In my opinion this phrase it needs to be rewritten because as it is, it seems confusing. ... rewrite something like ... the experimental data showed activities of the venom on fibrinogen, causing degradation of Aα and Bβ chains within 3.5 hours post snakebite.
21- At line 186 please correct … Snake venom serin proteases (SVSPs) serine proteses …
22- At lines 192 and 193 the authors wrote …. A major part of TLEs is formed by zinc metalloproteinases that preferentially cleave Aα chain of fibrinogen. Thrombin-like enzymes are serine proteases and zinc metalloproteases are other class of proteolytic enzymes classified as metzincins. Then please in the revised version do the correction.
23- At line 209 and through out of the text please substitute D-dim by D-dimer … The only issue remaining is the increase in the concentration of D-dim. D-dim
24- Regarding the materials and methods, the authors undoubtedly must change the descriptions of the methodologies used. Since SDS-PAGE and immuno Western-Blotting are the two main techniques used throughout the study, they need to be better described. As shown in this first version it is very summarized. It seems to me that the group has experience in the clinical area, but not in basic science. In the corrected version a more detailed description including protein dosage must be included and respective references. Something like this…Plasma protein samples were analyzed by a linear 4- 12% SDS-PAGE, purchased from ….., under reducing conditions, according Laemmli (1970). Plasma protein content was determined by the Coomassie Blue method (purchased from…….), as reported by Bradford (1976). For immunoblotting, (procedure was performed as described by Towbin et al.,1979), plasma proteins (… µg) were transferred to nitrocellulose (I suppose, but the authors did not described the membrane used) filters using Trans-Blot® SD (20V for …. min) (manufacturer …..). Fibrinogen immunodetection was performed using hyperimmune sera, and a secondary alkaline phosphatase-coupled anti-IgG purchased from (…….). The visualization of immunoreactions were done by using the BCIP/NBT substrate reaction (manufacturer………). Control of primary antibodies specificity was performed with pre-immune serum.
25- Also, in my opinion, the authors should dilute the blood plasma samples in some appropriate buffer such as PBS or saline and not simply water. The absence of ions can cause the formation of precipitates of some proteins and this can alter the electrophoretic run or interpretation of results.
26- It seems to me without a doubt that fibrinogen profiles change according to exposures to different venoms. However, the conclusions made by the authors that the fibrinogen chains are in fact degraded and the fragments generated, in my opinion, need to be confirmed by experiments using purified fibrinogen, because as already written above, human plasma has many proteins that colocalize in areas of the cut or generated fragments and this masks the interpretations.
Author Response
Reviewer No. 3:
General Comments:
In the manuscript submitted to TOXINS, the authors studied changes in plasma fibrinogen in patients who suffered accidents after being bitten by five different types of snakes (Genus Crotalus, Echis, Macrovipera, Atheris and Calloselasma). The analyzes were performed on the patients' plasma by means of Western Blotting reactions with an antibody that recognizes fibrinogen, and the authors reported that patients injured by snake bites of Crotalus, Echis, Macrovipera had degraded fibrinogen chains, while patients injured by Atheris and Calloselasma snakebites showed less significant fibrinogen degradation. The subject is consistent with the TOXINS line of publications, the text is clear and coherent, the authors have clearly shown their results and in my opinion is that a revised version with minor revision can be accepted for publication. The topic has a great impact on public health in developing countries, where many snakebite accidents are reported daily and these data can help in understanding the mechanistic of post-envenoming pathology.
Comments
- Between lines 8 and 9 the authors wrote …. we prospectively analyzed changes in plasma fibrinogen caused by the envenoming of five non European Viperidae snakes in six patients. Undoubtedly, the statistics of studying only 6 injured patients is low. This is a criticism that authors should be prepared to respond to! If a larger number of cases were studied, would the results always agree with the data described here?
A: This is a pilot study to investigate the potential of WB type of FBG analysis, the plasma fibrinogenolysis profile of envenomed patients. In European conditions with a small number of injured venomous snake breeders, the study could not be extended, reproducibility and statistical processing are not possible. Further studies ideally in collaboration with other Toxinology Centers in Europe and worldwide are needed. We added this text to the Limitations section of the manuscript.
- Between lines 22 and 24 the authors wrote …The envenoming by snakebites of most non-European viperids (family Viperidae, sub- family Viperinae and Crotalinae) causes, among others, venom-induced consumption coagulopathy (VICC). Although the authors are studying venom-induced consumption coagulopathy effects after accidents caused by different snakes, in this part of the text they could indicate the other types of clinical effects also described after envenomations! Making it clear that coagulation effects are not the only ones seen after accidents. This can undoubtedly help less experienced clinicians in the field who may have access to the text.
A: Thank you for this comment. More frequent clinical symptoms associated with the snake bite were added to the manuscript.
„The envenoming by snakebites of most non-European viperids (family Viperidae, subfamily Viperinae and Crotalinae) causes, among others (e.g., gastrointestinal disturbances, local edema and tissue damage, hypotension up to shock, kidney injury), venom-induced consumption coagulopathy (VICC). This type of coagulopathy decreases plasma fibrinogen (FBG) levels significantly, with fibrin/fibrinogen degradation products (FDP) occurrence. VICC may finally lead to bleeding or thrombotic microangiopathy and rarely to thrombotic complications [1, 2, 3].“
- Between lines 28 and 29 the authors wrote …Defibrination in VICC is caused by fibrino(geno)lytic enzymes including thrombin- like enzymes. In my opinion, the authors should replace Thrombin-like enzymes by proteolytic enzymes, since some metalloproteases can also degrade plasma substrates, in addition to serine proteases. It would also be more elegant to cite bibliographic references for this sentence!
A: Thank you for your comment. The text was corrected as recommended and additional references were added.
- Between lines 38 and 40 the authors wrote …A detailed description of the enzyme properties and their effect on coagulation factors “in vitro” in different snakes has been published elsewhere, mainly with the emphasis on the analysis of venom components. Here, no doubt, the authors could indicate some of these references. This would make the text more elegant and complete, in addition to showing that they (the authors have expertise in the literature of the area)!
A: Thank you for the suggestion to familiarize readers with some analysis describing the venom components affecting hemocoagulation. Few references were added to the manuscript.
Kini, R.M.; Rao, V.S.; Joseph, J.S. Procoagulant proteins from snake venoms. Haemostasis 2001, 31, 218–224. DOI: 10.1159/000048066
Swenson, S.; Markland, F.S. Jr. Snake venom fibrin(ogen)olytic enzymes. Toxicon 2005, 45, 1021–1039. DOI: 10.1016/j.toxicon.2005.02.027
Senis, Y.A.; Kim, P.Y.; Fuller, G.L.; et al. Isolation and characterization of cotiaractivase, a novel low molecular weight prothrombin activator from the venom of Bothrops cotiara. Biochim Biophys Acta 2006, 1764, 863–871. DOI: 10.1016/j.bbapap.2006.03.004
Wang, H.; Chen, X.; Zhou, M.; Wang, L.; Chen, T.; Shaw, C. Molecular Characterization of Three Novel Phospholipase A2 Proteins from the Venom of Atheris chlorechis, Atheris nitschei and Atheris squamigera. Toxins 2016, 8, 168. DOI: 10.3390/toxins8060168
5- Throughout the text, the authors describe their results obtained through WB reactions, analyzing the
plasmas of different patients with antibodies that recognize fibrinogen. The WB reactions, after standardized by the laboratory, are easy to reproduce and reasonably cheap. However, the authors could have described other inexpensive laboratory and clinical parameters, considering developing countries, such as blood count with platelet counts, physical examinations of patients performed by clinicians with experience in the area, seeking a correlation with the data obtained in the analyzes of plasma fibrinogen and possible streamlining of patient treatments. This could make the described data on fibrinogen degradation after envenomation more complete and attractive to readers in the field.
A: In addition to the tables, the limit values of PLT were added to the text. The main substitution therapy is also shown in the tables. The aim of the manuscript was to present mainly hemocoagulation findings of the patients, while detailed description of the clinical course was not the purpose of the study.
6- Between lines 61 and 62 the authors wrote … To monitor the effect of various snake venoms on the fibrinogen molecule, we used SDS-PAGE followed by western blot analysis of reduced patient plasma samples. Better rewrite this sentence … To monitor the effect of various snake venoms on the fibrinogen, we studied plasma from different patients submitted to SDS-PAGE under reduced conditions and Western Blotting reactions using specific antibodies.
A: Thank you for your comment, the sentence has been changed.
To monitor the effect of various snake venoms on the fibrinogen, we studied plasma from different patients submitted to SDS-PAGE under reduced conditions and Western Blotting reactions using specific antibodies.
7- Throughout the text, please substitute molecular weight for molecular mass. Molecules have molecular mass, not molecular weight. Who has weight is rice, bread, meat …. Although this term molecular weight is accepted and used in the literature, the most correct is molecular mass. There is no weight spectrometry, but mass spectrometry!
A: Thank you for your recommendation, the weight was changed to mass.
8- Between lines 64 and 66 the authors wrote... The intensity of the Aα chain bands and especially the γ chain band was visibly increased with prolonging time since snakebite. In my opinion, this diffuse band, with the shape of a fingernail facing downwards, and which is just below the fibrinogen Aα band, seems to be albumin, the most abundant protein in human plasma, and which colocalizes in this region on the SDS-PAGE under reduced conditions (Between 60-67 kDa). As albumin is very abundant in plasma, it gives the impression of a reaction with the anti-fibrinogen antibody, but it appears only due to the high concentration. As a control, the authors should use plasma obtained from uninjured patient and normalized in protein concentration as a reference, and they could see that this band in 67 kDa is albumin using an antibody for instance. It is also likely that the Aα chain has suffered a small degradation, and the product formed after the action of venom proteases now colocalize with Albumin. Giving the impression that the Aα chain has increased, but in reality it has decreased its molecular mass because it has been degraded.
A: Thank you for your comment. We admit that the fibrinogen zone could be deformed by the presence of albumin. During method optimization, proteins separated by SDS-PAGE were visualized using Coomassie Blue (CB) staining in each patient's plasma samples. The intensities of the albumin CB stained zones were comparable at all time points. In addition, we relied on the specificity of the chosen western blot method and the antibody against human fibrinogen used for the experiments.
9- Between lines 98 and 99, for SDS-PAGE WB reaction, figure 1a, why lanes started with 3, 4, 5 and 6 instead 1, 2, 3, ….
A: The samples No 1 and 2 in Table 1 were not analyzed by WB, so figures of WB analysis started by sample No 3 indicated in Table 1.
10- If the authors' objective is really to verify the action of toxins with proteolytic activities found in the crude venoms of different snakes, on fibrinogen, why didn't they study the actions of these crude venoms on purified fibrinogen, rather than on patient plasma? which is a very “contaminated” sample, containing hundreds of other proteins in addition to fibrinogen, and this masks the interpretations of the results.
A: This idea would be very interesting, indeed, but the aim of this pilot study was mainly to compare in a timeline the process and mode of FBG degradation in the context of the changes of other parameters of hemocoagulation in real patients. The findings of this pilot study may lead to further laboratory and experimental „in vitro“ studies.
11- Undoubtedly, the results postulated by the authors of plasma fibrinogen degradation by different snake venoms could be confirmed by the exposure of purified fibrinogen to different venoms. By doing this experiment, the authors could verify which fibrinogen chains are actually degraded by the different venoms and which peptides are generated after fibrinogen degradation. Human plasma has so many proteins that difficult the interpretation of results.
A: We agree, we added this to the limitation section of the manuscript.
12- In figures 1 and 2, the authors should have followed the same times after the different accidents, for a better comparative analysis. They Need to explain in the revised version, why times were different.
A: Blood sampling for the evaluation of hemocoagulation parameters were the part of routine blood sampling in the ICU, including the times of expected changes following therapeutic interventions, such as antivenom administration or application of blood products. Patients were not, in agreement with the Ethical Committee recommendation, bothered by additional experimental blood samplings. Furthermore, the patients were also admitted to the unit in different times since snakebite.
13- Between lines 71 and 73 the authors wrote about figure 1b …. In the case of Calloselasma rhodostoma envenoming, we found that intensity of FBG Aα and Bβ chains bands was decreased in the patient´s plasma samples 8.5 h after snake bite. In my opinion since there are colocalization of Aα chain and albumin this conclusion is masquerade and the authors can revise this conclusion performing this same experiment with purified fibrinogen!
A: Thank you for your interesting idea. It would certainly be an interesting extension for further work, our presented work is a pilot study as is mentioned in introduction. As for colocalization of Aα chain and albumin, we based our results on control staining during method optimization and specificity of used antibody against human fibrinogen.
14- Also regarding fig. 1b …. FBG fragments 27kDa and 44kDa appeared at all time points after snakebite in comparison with the human fibrinogen standards. The best comparative analysis in this case was with human plasma of a reference normal donor.
A: Thank you for your valuable suggestion. It would certainly be an interesting extension for further work. By choosing the FBG standard, we tried to ensure the clarity of monitoring the resulting FBG fragments.
15- In the lines 88 and 89, about figure 1c, the authors wrote … FBG fragments in 100- 140kDa … regions appeared. In a revised version better correct to fragments with approximately 120 and 150 kDa. The authors also need to explain why these same fragments are also present in the FBG used as a reaction control.
A: Thank you for your comment. FBG fragments were corrected. Purified human fibrinogen was used as a standard and it can covalently cross-link with other plasma proteins as reported by Mosesson et al. (Mosesson MW, Holyst T, Hernandez I, Siebenlist KR. Evidence for covalent linkage between some plasma a2-antiplasmin molecules and Aa chains of circulating fibrinogen. J Thromb Haemost 2013; 11: 995–8).
16- The WB reaction that describes the immunostaining of fibrinogen in figure 2a is questionable, since even the staining profile of the FBG used as a reference is not good! In this case the conclusions are questioned by the fact that even the control shows practically no marking of the γ chain. In addition, the Aα chain marking profiles also oscillate between lower markings in lanes 1 and 3 and higher markings in lanes 2, 4 and 5.
A: Thank you for your observation. Intensity of the γ chain bands were compared not only with respect to the standard, but also by comparing individual samples over time. The intensity of Aα chain bands was probably caused by course of treatment, concretely antivenom application. The intensity of these bands increased over time also as the effect of the wenom wore off.
17- All tables must be accompanied by legends explaining the respective reading times for PT/INR, would it be seconds or minutes?, would the letters (s) after APTT and TT be seconds? Also standardize the writing in the PT/INR or INR/PT in all tables
A: Legends with abbreviations were added to each Table.
Abbreviations: PT/INR, prothrombin time/the international normalized ratio; APTT, activated partial thromboplastin time; TT, thrombin time; s, seconds; AT III, antithrombin III; FBG, fibrinogen; PLT, platelets; NA, not applicable.
18- It is good procedure that morphological descriptions have symbols indicating the alterations shown. Thus, in the Figure 3 of the revised version, the authors could place an arrow pointing to the hemorrhagic blister in the injured patient's hand. This may seem very obvious, but it follows a good procedure of morphological and descriptive analysis!
A: The arrow has been placed to Figure 3 in the manuscript.
19- At line 176, … The most significant changes in western blot analysis under reduced conditions …change to … The most significant changes in western blot analysis followed a SDS-PAGE under reduced conditions… because electrophoresis that was under reduced conditions and not WB!
A: Thank you for your comment, the sentence has been changed.
„The most significant changes in western blot analysis followed a SDS-PAGE under reduced conditions“.
20- At lines 178 to 181 the authors wrote ….In these cases, we found no visible, minimally visible, or significantly decreased intensity of FBG chains bands. In the first case of Crotalus adamanteus envenoming, both FBG Aα and Bβ chains bands were not visible 3.5 h post snakebite. In my opinion this phrase it needs to be rewritten because as it is, it seems confusing. ... rewrite something like ... the experimental data showed activities of the venom on fibrinogen, causing degradation of Aα and Bβ chains within 3.5 hours post snakebite.
A: Thank you for your recommendation, the sentence has been changed.
„The experimental data showed activities of the venom on fibrinogen, causing degradation of Aα and Bβ chains within 3.5 hours post snakebite.“
21- At line 186 please correct … Snake venom serin proteases (SVSPs) serine proteses …
A: corrected – “serine proteases”
22- At lines 192 and 193 the authors wrote …. A major part of TLEs is formed by zinc metalloproteinases that preferentially cleave Aα chain of fibrinogen. Thrombin-like enzymes are serine proteases and zinc metalloproteases are other class of proteolytic enzymes classified as metzincins. Then please in the revised version do the correction.
A: Thank you for your comment, the text was corrected.
23- At line 209 and through out of the text please substitute D-dim by D-dimer … The only issue remaining is the increase in the concentration of D-dim. D-dim
A: The abbreviation D-dim was changed to full description D-dimer.
24- Regarding the materials and methods, the authors undoubtedly must change the descriptions of the methodologies used. Since SDS-PAGE and immuno Western-Blotting are the two main techniques used throughout the study, they need to be better described. As shown in this first version it is very summarized. It seems to me that the group has experience in the clinical area, but not in basic science. In the corrected version a more detailed description including protein dosage must be included and respective references. Something like this…Plasma protein samples were analyzed by a linear 4- 12% SDS-PAGE, purchased from ….., under reducing conditions, according Laemmli (1970). Plasma protein content was determined by the Coomassie Blue method (purchased from…….), as reported by Bradford (1976). For immunoblotting, (procedure was performed as described by Towbin et al.,1979), plasma proteins (… µg) were transferred to nitrocellulose (I suppose, but the authors did not described the membrane used) filters using Trans-Blot® SD (20V for …. min) (manufacturer …..). Fibrinogen immunodetection was performed using hyperimmune sera, and a secondary alkaline phosphatase-coupled anti-IgG purchased from (…….). The visualization of immunoreactions were done by using the BCIP/NBT substrate reaction (manufacturer………). Control of primary antibodies specificity was performed with pre-immune serum.
A: Thank you for your comments, we have modified the text according to your suggestions.
25- Also, in my opinion, the authors should dilute the blood plasma samples in some appropriate buffer such as PBS or saline and not simply water. The absence of ions can cause the formation of precipitates of some proteins and this can alter the electrophoretic run or interpretation of results.
A: We thank the reviewer for pointing out this mistake, which has been corrected.
Samples were first diluted with PBS and then diluted with water according to the sample preparation protocol for NuPAGE Bis-Tris Mini Gels.
26- It seems to me without a doubt that fibrinogen profiles change according to exposures to different venoms. However, the conclusions made by the authors that the fibrinogen chains are in fact degraded and the fragments generated, in my opinion, need to be confirmed by experiments using purified fibrinogen, because as already written above, human plasma has many proteins that colocalize in areas of the cut or generated fragments and this masks the interpretations.
A: This idea would be very interesting, indeed, but the aim of this pilot study was mainly to compare in a timeline the process and mode of FBG degradation in the context of the changes of other parameters of hemocoagulation in real patients. The findings of this pilot study may lead to further laboratory and experimental „in vitro“ studies.
We agree with the reviewer that the changes seen in our pilot cohort should be confirmed by experiments with the use of purified fibrinogen and we put this fact to the Limitations section of revised manuscript.
Reviewer 4 Report
After careful revision of the manuscript it was found that the manuscript covers an important topic and worthy of study and can be published after minor revision which does not affect the quality of the manuscript. I believe that
1) A deeper statistical analysis could enrich this paper.
2) some references could enrich this manuscript, for instance:
Current Protein and Peptide Science, Volume 20, Number 5, 2019, pp. 471-487(17)
Phytotherapy Research , Volume 10, 1996, pp. 58-61
Author Response
Reviewer No. 4:
After careful revision of the manuscript it was found that the manuscript covers an important topic and worthy of study and can be published after minor revision which does not affect the quality of the manuscript. I believe that
- A deeper statistical analysis could enrich this paper.
A: We thank the reviewer for this comment but deeper statistical analysis in this pilot study is not possible due to low number of probands. This is a pilot study to investigate the potential of WB type of FBG analysis, the plasma fibrinogenolysis profile of envenomed patients. In European conditions with a small number of injured venomous snake breeders, the study could not be extended, reproducibility and statistical processing are not possible. Further studies are needed. We added this text to the Limitations section of the manuscript.
2) some references could enrich this manuscript, for instance:
Current Protein and Peptide Science, Volume 20, Number 5, 2019, pp. 471-487(17)
Phytotherapy Research , Volume 10, 1996, pp. 58-61
A: We would like to thank the reviewer for this interesting recommendation. However, we feel that these articles are not very much related to the studied problem of FBG degradation in VICC.
Round 2
Reviewer 2 Report
The authors amend the treatise one by one in response to the many points pointed out by the reviewers. As a reviewer, I would like to accept this revised version.
Reviewer 3 Report
Dear Toxins/MDPI Researchers and Editors. After careful reading of the revised manuscript and the letter responding to my suggestions regarding the original manuscript, it is my opinion that the text has been greatly improved and can now be accepted for publication in TOXINS! The authors made several modifications throughout the revised text, which made the manuscript more elegant and attractive. They also answered all my questions clearly and convincingly.